# A systematic review of qualitative research on the physical and mental health impacts of immigration detention on asylum seekers and refugees

Bafreen Sherif[1,2]*, Debbie C. Hocking[1], Susan Rees[3], Letizia M. Affaticati[1,2,4], Suresh Sundram[1,2,5]*

1 Department of Psychiatry, School of Clinical Sciences, Faculty of Medicine, Nursing and Health Sciences, Monash University, Clayton, Melbourne, Victoria, Australia, 2 Cabrini Outreach, Asylum Seeker and Refugee Health Hub, Melbourne, Victoria, Australia, 3 Department of Psychiatry and Mental Health, School of Clinical Medicine, Faculty of Medicine and Health, University of New South Wales, Sydney, New South Wales, Australia, 4 Department of Medicine and Surgery, University of Milano Bicocca, Monza, Italy, 5 Monash Health, Monash Medical Centre, Clayton, Melbourne, Victoria, Australia

* Bafreen.sherif@monash.edu (BS); suresh.sundram@monash.edu (SS)

## Abstract

Globally, the number of refugees and asylum seekers has reached unprecedented levels, prompting many host countries to adopt immigration detention as a strategy to deter unauthorised arrivals. While previous studies have largely used quantitative methods to examine the consequences of detention, this is the first qualitative systematic review synthesising evidence on the impact of immigration detention on the mental and physical health of adult and child asylum seekers and refugees (ASR) in middle- to high-income host countries. A systematic search of medical, allied health, and social science databases and grey literature was conducted between December 2021 and October 2024 (Embase, MEDLINE, PsycINFO, CINAHL Plus, Global Health, APA PsycARTICLES, Social Policy and Practice, Cochrane Library, Scopus, and Open Grey). Reference lists of included studies were also screened. The review was registered with PROSPERO (CRD42022328867). Study selection, critical appraisal, and data extraction were performed using the Joanna Briggs Institute Critical Appraisal Checklist for Qualitative Research. Of 2,512 records identified, 564 full texts were assessed, and 20 studies met the inclusion criteria, comprising 374 adults and 139 children with current or past detention experiences, with detention durations ranging from two weeks to 38 months. Thematic synthesis revealed five overarching themes: human rights violations within detention; systemic failures of healthcare provision and resultant vulnerability; adverse health impacts of detention; changes in self-concept, relationships, and worldview after release; and enduring negative consequences for mental and physical health. Findings highlight that immigration detention uniformly undermines the health and well-being of ASR, with effects persisting

**Data availability statement:** This submission contains all data required to replicate the findings of the review. The minimal dataset consists of the search strategy, screening and selection process, and tables of included studies with extracted data, all of which are provided in the manuscript and associated files. As this study is a systematic review of previously published qualitative research, no new primary data were collected. Full transcripts or raw data from the original studies are not available to us, as they remain the property of the original authors.

**Funding:** The authors received no specific funding for this work.

**Competing interests:** The authors have declared that no competing interests exist.

long after release. This review underscores the urgent need for health and human rights to be central considerations in migration policy and practice.

---

## Author summary

### Why was this study done?

- Globally, the population of refugees and asylum seekers (ASR) has attained unprecedented levels, with some host countries resorting to immigration detention as a strategy to curb an influx of unauthorised arrivals.

- This is the first systematic review investigating qualitative evidence related to the mental and physical health of asylum seekers and refugees who have experienced current or past immigration detention in middle- to high-income host countries.

### What did the researchers do and find?

- We performed a comprehensive literature search examining qualitative evidence related to the mental and physical health of adult and child ASR with a current or past experience of immigration detention in middle-to-high-income host countries.

- In accordance with PRISMA, SRQR, and JBI guidelines, we implemented a rigorous and transparent methodology to facilitate replication of the systematic review process and strengthen the reliability of our results.

- Despite the diverse qualitative methodologies employed in the included studies, coherent and integrated findings were extracted, facilitating unique insights that had previously been unexplored.

- This systematic review and thematic synthesis highlight the substantial long-term effects that immigration detention inflicts on the psychological well-being of both adult and child ASR, both during their period of detention and aftermath. Additionally, it advocates for further research to bridge the current gap in evidence regarding the physical health of refugees and asylum seekers who have encountered immigration detention.

### What do these findings mean?

- There is a pressing need for additional research focused on the development and evaluation of interventions designed to support ASR in navigating the substantial changes in their lives following the experience of immigration detention, with particular attention to their mental and physical health.

- The findings derived from this synthesis hold significant potential for health professionals and policymakers involved in immigration detention, as they may aid in guiding the future formulation of treatment protocols specifically designed to address the needs of refugees and asylum seekers.

- The predominant body of research has focused on forced migrant populations in the post-detention context, rather than recruiting participants during their detention period. This methodological approach has the potential to introduce recall bias in their narratives and the ensuing synthesis of data.

## 1. Background

Unprecedented flows of people seeking international protection have led many industrialised countries to introduce increasingly restrictive policies aimed at curtailing entry [1,2]. One such strategy is the use of immigration detention to deter entry and facilitate the expulsion of those deemed to have entered irregularly or without authorisation [3–5]. Once apprehended, immigration detainees find themselves subjected to conditions that may have a detrimental effect on their overall health, and which, in extreme cases, may lead to mortality [6]. Detainees may be subjected to verbal and physical maltreatment, family separation, restraints, solitary confinement, inadequate nutrition, overcrowding, poor sanitation, and poor air and water quality, among other substandard living conditions [2,7–11].

In the immigration context, detention centres are facilities used to confine individuals—typically asylum seekers, refugees, or irregular migrants—pending the outcome of their immigration case or removal from the host country. They are widely characterised as prison-like environments, often with limited personal space, high-security fencing, and constant surveillance by guards, which restrict autonomy and opportunities for meaningful activity [12–14]. Daily routines are heavily regulated by policy [12,15], and the environment is often associated with a fear of deportation, distrust of host communities, and the generation of persistent stress [16]. Broader effects include reduced healthcare usage [17,18], mistrust of authorities, and wariness in engaging with institutions [19,20]. In some cases, fear of re-detention or deportation during healthcare interactions has led detainees to view health services as a form of "biopolitical surveillance" [21].

Given the variation in detention systems across countries and the differing terminology associated with detainees in various studies, this review draws on definitions from the United Nations High Commissioner for Refugees (UNHCR) and the International Organisation for Migration (IOM) to clarify key terms. According to the UNHCR, a **refugee** is someone who, "owing to a well-founded fear of being persecuted for reasons of race, religion, nationality, membership of a particular social group or political opinion, is outside their country of nationality and is unable or unwilling to avail themselves of the protection of that country," as recognised under the 1951 Convention Relating to the Status of Refugees and its 1967 Protocol. Refugee status is conferred following a formal determination process. Although it is a contravention of the Convention Relating to the Status of Refugees for signatory nations to detain refugees, refugees from other nations may still be detained. An **asylum seeker** is someone who has applied for recognition as a refugee and is awaiting a decision on their claim. Asylum seekers may present themselves at a border or already be within the territory of the receiving country [22]. They are entitled to remain in the country of application until a decision is made regarding their application.

This review focuses specifically on **asylum seekers and refugees (ASR)** who are held in immigration detention facilities, whether at borders, offshore sites, or within the territory of the host country. We use the collective term "ASR" throughout to reflect both the legal distinctions and the frequent interchangeability of these terms in academic and policy discourse, while acknowledging the lack of consistent usage across the literature.

### 1.1. The influence of immigration detention on physical and mental health outcomes

#### i) Adults

ASR populations experience higher levels of post-traumatic stress disorder (PTSD), anxiety and major depressive disorder (MDD) compared with the general population [23–26]. The detention environment may further aggravate mental disorders, as indicated by elevated rates of both physical and psychological health conditions, self-harm, and suicide attempts among immigration detainees [27,28]. This is particularly notable among individuals who have encountered pre- and peri-migration trauma and persecution [29–31].

The structural mechanisms of immigration detention, including inadequate medical care and substandard living conditions, may compromise the health of detainees and create obstacles to accessing necessary medical attention [3,9,27,30,32–36]. The detention environment has also been associated with physical health problems in detainees, such as gastrointestinal issues, respiratory infections and musculoskeletal complaints [37]. Access to sufficient healthcare and medical services within detention facilities may lead to untreated or poorly managed medical conditions, culminating in deteriorated physical health outcomes [37]. Furthermore, individuals may continue to experience adverse health consequences even after their release from detention [3,38,39]. Detainees frequently endure persistent symptoms of mental disorders such as MDD, PTSD, and anxiety, which may impede their successful reintegration into society [3,38,39]. Following release from detention, social and economic challenges can further erode overall health and well-being [40].

### ii) Children

Similar to adults, immigration detention has been found to negatively affect the physical and mental health of children, with deterioration in mental health directly linked to the cumulative exposure to stress and trauma [41]. Immigration detention is recognised as a significant adverse childhood experience, which contributes to or worsens negative health outcomes among children and adolescents who arrive as asylum seekers [42–48]. Additionally, immigration detention acts as a post-migration stressor, significantly increasing the prevalence of Post-Traumatic Stress Disorder (PTSD) and Major Depressive Disorder (MDD) in children [41,49,50]. In addition to the mental and physical health consequences, immigration detention is associated with adverse neurodevelopmental, educational, and family impacts [10,41–45].

### 1.2. Literature gaps

The prominent systematic reviews within this domain have utilised quantitative data to examine variances in symptom endorsement patterns, specifically anxiety, depression, and post-traumatic stress disorder among detained and non-detained adult or child refugees [27,28,42,46–48]. Additional systematic reviews have further incorporated aspects concerning physical health and social functioning among confined asylum seekers in detention [46,49], alongside identifying distinct detainee characteristics correlated with an elevated risk within immigration detention [50]. These characteristics include the duration of detention, pre-existing trauma, such as torture and sexual violence [51], pre-existing mental and physical health issues, and the inadequate healthcare and mental health services available within the immigration detention system. The existing reviews of qualitative studies have assessed related subjects, such as the healthcare needs of migrants while detained in European immigration detention environments [52] and a meta-ethnographic approach evaluating the effects of immigration detention on the health and well-being of ASR [53].

To the best of the authors' knowledge, no systematic reviews have been conducted on the physical and mental health impacts of immigration detention on ASR, using qualitative studies. Thus far, global research pertaining to the physical and psychological health of ASR who have experienced immigration detention has predominantly employed cross-culturally validated quantitative assessment tools. Nevertheless, it is conceivable that certain attributes of the immigration detention experience, along with its corresponding health effects, may not be sufficiently captured within the current diagnostic criteria or measurement instruments. To investigate this potential gap, we undertook a systematic review of qualitative studies that examine the physical and mental health of ASR populations within the framework of their immigration detention experience.

### 1.3. Aim and significance of this review

This systematic review was conducted to explore the qualitative literature describing the impact of immigration detention on the physical and mental health of adult and child ASR in middle- to high-income host countries. This is the first qualitative systematic review to assess physical and mental health associations with current/ prior detention in both adult and child populations.

## 2. Methods

### 2.1. Protocol and eligibility criteria

Systematic searches for peer-reviewed qualitative literature were conducted utilising the Joanna Briggs Institute (JBI) Checklist for Qualitative Research [54]. Studies were included if they met recognised qualitative reporting standards (Standards for Reporting Qualitative Research, SRQR) [55]. All authors (BS, DH, LA, SS, SR) contributed to the systematic review process. Title and abstract screening were performed independently by two authors (BS and DH), with disagreements resolved through consensus with the additional authors (LMA, SR and SS). Data extraction was carried out independently by two reviewers (BS and DH) and then cross-checked by the remaining team (LMA, SS, SR). BS performed the data analysis, and data interpretation was undertaken collaboratively by BS, DH, SS, and SR. The protocol for this systematic review was registered with the International Prospective Register of Systematic Reviews (PROSPERO), registration number CRD42022328867.

### 2.2. Inclusion and exclusion criteria

Qualitative or mixed-methods studies were included if they i) sampled adults or children detained for immigration purposes, either whilst detained or post-release from immigration detention, and ii) described physical and mental health outcomes as a consequence of immigration detention.

Studies were excluded on the following grounds: i) did not report specifically on the health impact of immigration detention; ii) detention exclusively had a criminal justice purpose; and iii) detention did not deprive the freedom of movement (e.g., migrant reception centres).

Qualitative or mixed-methods studies of ASR, as well as observations of key informants, professionals, or stakeholders working with ASR, were included. No restrictions were placed on age, sex, detainee country of origin, or method of describing physical and psychological health outcomes. No restrictions were placed on the publication date or language of publication.

### 2.3. Information sources and search strategy

Relevant studies were identified through electronic searches of Embase (1980–2024 week 43), OVID MEDLINE (1946–2024 week 43), APA PsychINFO (1806–2024 week 43), CINAHL Plus (1937–2024 week 43), OVID platform (Global Health (1910–2024 week 43), APA PsycARTICLES (1967–2024 week 43), Social policy and practice (from 1981 to 2024 week 43), Cochrane Library (from 1993 to 2024 week 43), Scopus, Open Grey and advanced Google search of.org websites until ten consecutive pages did not include relevant pages (searched 2024 week 29). A comprehensive search strategy was developed for Ovid Medline and then modified for other databases, utilising an earlier review by von Werthern et al. (2018) [56] as a template (S1 Text). Electronic searches were supplemented with the screening of reference lists of included primary studies. All study titles and abstracts were imported into an Excel spreadsheet and stored by the primary author.

### 2.4. Study selection and data collection

Study title and abstract screening, as well as full-text screening, were conducted by the primary author against predetermined criteria, with three additional reviewers (DH, LMA, SS) providing input when any disagreements arose. Relevant data were extracted and entered into an Excel sheet by the primary author, who then accuracy-checked and consolidated the data independently with three reviewers (DH, LMA, SS). A fourth reviewer (SR) provided overarching support and consolidation. We extracted data pertaining to general study characteristics and methodology, participant demographics, and qualitative content, as well as detention duration, where stipulated, which is described in detail in S1 and S2 Data.

## 2.5. Quality Assessment

The quality appraisal of the included studies was conducted using the Joanna Briggs Institute (JBI) Critical Appraisal Checklist for Qualitative Research (S1 Table) [54]. All studies were retained in the review regardless of their methodological quality, consistent with established approaches to qualitative evidence synthesis [57]. Excluding studies based on methodological quality would have further restricted the already limited body of literature meeting our inclusion criteria, thereby hindering the provision of a comprehensive overview of the field. This decision reflects the view that participant insights remain valuable even when studies have certain methodological limitations, enabling a richer understanding of the experiences of ASR. The results of the quality appraisal were considered in the interpretation of findings, and methodological limitations were explicitly acknowledged in the synthesis and discussion to ensure transparency regarding the overall strength and trustworthiness of the evidence base [57].

## 2.6. Summary measures and synthesis of results

Results were analysed using a textual narrative synthesis approach, as described by Lucas et al. [58], in which individual study characteristics and findings are reported according to a standard format, and structured summaries are developed to elaborate on and contextualise the data extracted across studies [58,59]. The qualitative context extracted included child or adult ASR participants or observations of key informants, professionals or stakeholders working with ASR either during or in the aftermath of the immigration detention experience and described the physical and/or mental health outcomes as a consequence of immigration detention. Qualitative data extraction was performed by the primary author (BS) and entered into an Excel spreadsheet. Following initial data extraction, synthesis of findings from individual studies was done by (BS) with input from study authors (DH, LA, SS, SR), focusing on the scope, differences and similarities among studies to make broader inferences about the impact of immigration detention on the physical and mental health outcomes of adult or child ASR [58]. Data domains were grouped in relevance across respective themes, which were agreed upon by the study researchers.

## 3. Results

### 3.1. Search results

Out of 2512 individual records identified, 20 studies were eligible for inclusion – Fig 1. Of the 20 studies included, seven were conducted in Australia [12,14,15,60–63], four in the United Kingdom [64–67], three in Canada [68–70], three in the USA [2,21,71], one in Serbia [72], one in Sweden [13] and one in the Netherlands [73]. Of the studies, six were published in the last five years. S1 Data presents the characteristics of the included studies. Studies were conducted in high-income countries, with only 1 study reported in a middle-income country.

### 3.2. Study characteristics

Study characteristics are summarised in S1 Data. A total of 513 participants (374 adults and 139 children) were included across the studies, comprising refugees (N = 222), asylum seekers (N = 153), and a combined ASR group (N = 138). Participants originated from North Africa, Middle Africa, East Africa, West Africa, East Asia, Central Asia, South Asia, West Asia, Europe, North America, Central America, South America and the Caribbean. Fifteen studies focused exclusively on adult ASR [2,3,12–15,21,61,63–68,73], three on children and families [69,70,74] and one on adolescents [71]. One study focused on adult male undocumented migrants [72].

Three studies included observations of the health of ASR by health professionals [66], health and social service workers [12] and detention centre staff (mental health, welfare or religious services) (n = 11) [67]; two studies related to detainee self-harm and suicide in detention [21,67]; one study included four migrant women who had been held in detention while pregnant [66]. One study included 9 ASR patients who had been treated for PTSD at a specialist mental health

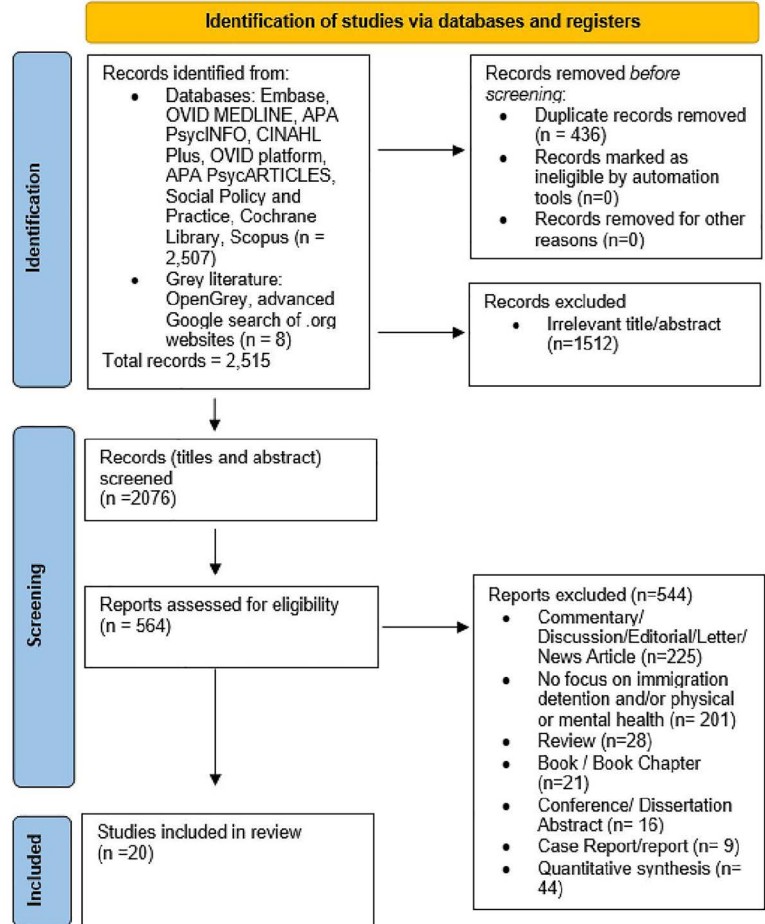

**Fig 1. PRISMA Flow diagram for the systematic review of qualitative studies.**

service and had previously been in immigration detention [73]. While all included studies addressed mental health, only two specifically reported on physical health issues [22,69].

### 3.3. Quality assessment

The JBI was used to characterise the strengths, limitations, and impact of pertinent data in addressing the research objectives and the quality of evidence across the included studies [54]. Study-level assessments are provided in S1 Table.

Among the 20 studies, nine met the criterion for congruence between their philosophical perspective and research method (criterion 1). Twelve studies did not state their philosophical or theoretical premises. Eighteen studies demonstrated alignment between the research methodology and the questions or objectives (criterion 2), and the same 18 studies aligned their methodology with data collection methods (criterion 3). Two papers lacked clarity on methodology. Seventeen studies demonstrated congruence between methodology and data analysis (criterion 4); this was unclear for one study due to poor description and for two due to insufficient detail. All 20 studies aligned methods with the interpretation of results (criterion 5). Only four studies reported the researcher's cultural or theoretical positioning (criterion 6), and two addressed the influence of the researcher on the research (criterion 7). In all studies, we determined that participants and their voices were adequately represented (criterion 8). Ethical approval by an appropriate body was reported in 15 studies (criterion 9). Three studies did

not provide such statements, and the criterion was not applicable to two studies due to the type of data sources utilised. Conclusions were appropriate for 19 papers; for one study, conclusions were deemed too partial (criterion 10).

## 3.4. Synthesis findings

Five major themes (Fig 2) were identified through the qualitative synthesis: three themes were relevant to the detention period (Themes 1–3), and two concerned the post-detention period (Themes 4–5). S2 Data references supporting quotes from the included studies within their respective themes.

## 3.5. Detention period

### 3.5.1. Theme 1 – Human rights violations in detention.
Five subthemes were identified within this theme: constriction of agency and autonomy, endemic uncertainty, systemic deprivation, dehumanisation and perceived injustice, and loss of safety.

The curtailment of individual agency and autonomy as a result of restrictions imposed while in detention culminated in the experience of imprisonment and removal of freedom. Authorities were viewed as omnipotent in their exercise of control and curtailment of detainee agency, with a resultant feeling of futility in seeking legal advice and powerlessness in being unable to influence one's life course [13]. ASR described threatening behaviour by the authorities and the resultant loss of safety and a lack of control over their own lives, substandard living conditions, and the absence of support systems to mitigate the stressful living conditions [13].

Witney et al. (2016) noted the endemic uncertainty and the sense of living 'in limbo' whilst waiting for asylum claims to be processed, with subsequent anxiety regarding the tenuous circumstances and concerns about forced repatriation [75]. Detention was described as a prison and, at times, was perceived to be worse than prison due to the uncertainty created by an information vacuum. Increased stress levels and a reduction in sleep quality were attributed to this endemic uncertainty, with life becoming more "miserable" as the length of detention extended [13]. Paradoxically, detainees believed they would be able to manage a longer period of detention if the exact duration were known to them [13]. Boerma et al. (2022) likewise noted that the endemic uncertainty arising from a deficiency in information and communication concerning their expectations of safety upon arrival in the Netherlands, coupled with the lack of clarity regarding detention procedures, exacerbated feelings of fear, anxiety and confusion [73]. Similar sentiments were observed in Hollis et al. (2019) [64], whereby asylum-seeking participants expressed that the inconsistent and inadequate communication they received from immigration authorities during their detention was a significant precursor to feelings of powerlessness and, consequently, a restriction of their agency.

Arshad et al. (2018) observed that the systemic deprivation characterised by neglect of fundamental needs and inadequate living conditions, including food scarcity and lack of privacy, significantly overshadowed and detracted from the pregnancy experiences of ASR women [66]. Similarly, participants in an Australian study by Passardi et al. (2022) described systemic deprivation, a lack of agency, dehumanisation, exposure to violence, and a loss of safety. Dehumanisation also encompassed feeling used for political gain to "stop the boats" without regard for the personal and community impact, with numerous individuals expressing regret about coming to Australia; some wished they had perished during the maritime journey [15]. Participants also felt confronted by the lack of autonomy over their lives; viewing deprivation as intentional, with guards either creating their own rules or appearing to act inhumanely, which further "killed [their] minds" [15].

The distress experienced by some ASR was attributed to living a confined and highly controlled existence, with limitations on self-determination and autonomy over the most basic aspects of daily life [12,68]. Constriction of agency and systemic deprivation were further evidenced by inadequate access to education and training, substandard sanitation, and confinement [12]. Additionally, restrictive routines, limited access to finances, and the inability to engage in work or social activities beyond the detention centre, coupled with high levels of monitoring and security personnel, further underscored the environment of constraint [12].

**Theme 1 – Human Rights Violations in Detention.**

i. Constriction of agency and autonomy
ii. Endemic uncertainty
iii. Systemic deprivation
iv. Dehumanisation and perceived injustice
v. Loss of safety

**Theme 2 – Systemic failures of healthcare provision and resultant ASR vulnerability and deprivation.**

i. Institutional and organisational failure
ii. Structural barriers to medical care

**Theme 3 – Negative health impacts of immigration detention**

i. Psychological health
ii. Physical health
iii. Impact on children

**Theme 4 – Changes in self-construct, relationships and worldview.**

• Fragmentation of identity and changes in self-perception
• Erosion of trust and social connectivity.

**Theme 5 – Enduring impact on mental and physical health**

• Continued psychological sequelae
• Physical health deterioration

**Fig 2 . Thematic Qualitative Synthesis.**

Even for young and healthy participants, systemic deprivation through neglect of basic human needs and experiences of dehumanisation significantly eroded well-being [2]. Loss of safety was further conveyed through accounts of multiple traumatic and violent experiences, including rape or the rape of a family member- either in the host community or by a detention guard- or threats of sexual violence [15].

Loss of safety was also prominent for children, who recounted adversity and violence experienced while in detention [70]. Play therapy reflected children's awareness of the lack of protection, even in the context of migration-related hopes for a better quality of life [70]. Similarly, Rothe et al. (2003) [71], reported that 74 adolescents detained at Guantanamo experienced loss of safety and psychological trauma through exposure to riots, violence, and suicide attempts by other detainees within the camp. Adolescent girls identified the absence of privacy and limited autonomy, in conjunction with threats of sexual assault from males within the camp, as their most significant and urgent concern. Juxtaposed to this, sexual promiscuity and risk of pregnancy increased the opportunity of expulsion from detention for adolescent girls. This phenomenon was theorised to be driven by a desire to terminate the experience of detention and, hence, the limitations on agency and autonomy [71]. Neglect of adolescents' basic needs was also apparent in the unsanitary living conditions of the Guantanamo detention camp, alongside a bland diet, which, in concert with the effects of heat and sensory overload, amplified tension and distress among the adolescent ASR. Adolescent participants reported feeling "treated like cattle," underscoring their experience of dehumanisation. In the chaotic conditions of detention confinement, the family unit was viewed as protective: a powerful anchor point offering safety, support and nurturance to the adolescent ASR [71].

Participants in Shishegar et al. (2021) also described loss of autonomy and agency in detention, linked to feelings of dehumanised and being treated like criminals [14]. Being confined in a prison-like environment was viewed as a violation and a departure from the expectations of safety and freedom upon arrival [14]. Johnston et al. (2009) found that perceived injustice and dehumanisation were key precipitants of anger—sometimes expressed outwardly through protests to assert rights and agency, and at other times directed inwardly, reinforcing the dominant narrative about being an ASR. This internalisation often involved feelings of being "undeserving" or believing detention was divine punishment for past wrongdoing, rather than a consequence of seeking safety and a new life [61].

Similarly, Diaz et al. (2023) noted ASR experienced enduring injustice associated with being detained in a prison-like environment, treated like criminals, and feeling 'more imprisoned than in prison', while living in a chronic state of uncertainty endemic to the asylum process [2]. Mares & Jureidini (2004) reported that children often perceived the refugee determination process as capricious, generating uncertainty and anxiety; they struggled to understand why some families they knew in detention had been granted visas while they had not. This apparent randomness fostered a sense of injustice, culminating in profound anger and worthlessness [62].

### 3.5.2. Theme 2 – Systemic Failures of Healthcare Provision and Resultant ASR Vulnerability and Deprivation.
Two subthemes were identified within this theme: institutional and organisational failure; and structural barriers to medical care.

Utilising qualitative data from Immigration & Customs Enforcement (ICE) inspection reports, contracts, and detainee death reports, Marquez et al. (2021) determined that suicide in immigration detention facilities was a by-product of institutional and organisational failure. Institutional failures, including deviations from care standards and inadequate staff training, were noted to contribute to a crisis of vulnerability and deprivation within detention systems [21]. Federally mandated protocols for detainee well-being were often not implemented in a timely manner. These systemic failings disproportionately affected minority populations, such as Latinx detainees, who are overrepresented in U.S. immigration detention. Between 2003 and 2015, 150 immigrant detainees died in ICE custody, with 13% of deaths due to suicide—an outcome attributed to insufficient suicide prevention training and intervention. This reflected a culture in which substandard care was normalised, producing inequitable health outcomes for marginalised groups [21].

In a similar context, Diaz et al. (2023) described structural barriers to accessing timely and adequate medical care, including delays and interruptions in medical services, insufficient access to mental health care, denial of language

interpreting services, inappropriate treatment and monitoring of COVID-19, and inadequate discharge planning. Furthermore, the majority of individuals experienced prolonged detention with restricted access to medical and mental health care, signifying a structural barrier. This situation constituted a significant impediment to addressing pre-existing health conditions and contributed to the emergence of new medical issues during the period of detention [2].

In the UK, Kellezi et al. (2016) studied detainees at Yarl's Wood Immigration Removal Centre (IRC). They found that 70% reported suicidal ideation, yet 78% of those who had self-harmed since arrival did not have an Assessment Care in Detention and Teamwork (ACDT) plan in place. ACDT plans are a structured system for monitoring and supporting at-risk detainees. Their absence represented a significant systemic failure with the potential to endanger vulnerable individuals. Some detainees feared disclosing mental health concerns or self-harm tendencies in case it adversely affected their immigration cases, despite the recognition of accessible mental health care as a basic human right. Even short periods of detention were perceived to harm physical and mental health. Several detainees believed that a formal health assessment would help identify emerging health issues. [67]

A UK study by Arshad et al. (2018) examining pregnant women in detention [66] illuminated the structural inadequacies in healthcare provision affecting an especially vulnerable population. The women in the study reported encountering significant barriers to medical care, including denial of access to timely and appropriate maternity services during detention, as well as a lack of attention from staff despite being notified of health concerns, further exemplifying systemic failures. Furthermore, the UK guidelines by the National Institute for Clinical Excellence (NICE) regarding the timely availability of antenatal care appointments [76] were not adhered to, resulting in a notable lack of continuity of care. One woman had prescribed regular medication withheld, resulting in suicidality and fear for the welfare of her baby. Additionally, external medical appointments were frequently cancelled due to insufficient detention guards available to escort the women, which represents both an institutional failure and a structural barrier to obtaining medical care. Ultimately, these factors illustrate a pervasive breakdown in healthcare provision, with some women reporting the deterioration of pre-existing mental health conditions that were aggravated and neglected by the detention experience [66]. In a comparable context, Hollis (2018) recounted the experience of a pregnant asylum seeker who suffered significant weight loss due to food deprivation in her accommodation, stemming from rigid adherence to regulations over basic human rights, adversely affecting both her health and that of her unborn child [64].

Hollis (2018) [64] also described structural barriers in instances where pre-existing health conditions, such as chronic back pain, were minimised during detention through insufficient pain management and refusal to refer patients to tertiary hospitals. This systemic failure in healthcare provision rendered a participant vulnerable by compounding his desolation in the face of perceived institutional indifference to his condition. Zimmerman (2012) [65] further illustrated systemic healthcare deficiencies in a case vignette of an asylum seeker who engaged in self-harm due to fears of refoulement. Despite sustaining significant leg and back injuries, the detainee perceived medical staff as lacking empathy, failing to listen to concerns, and inadequately addressing pain by prescribing only basic analgesics—exacerbating both vulnerability and emotional distress.

### 3.5.3 Theme 3 – Negative health Impacts of immigration Detention.

Three subthemes were identified within this theme: psychological health; physical health; and impact on children. Of note, the latter subtheme was delineated to reflect the distinct presentation of mental disorders, the differing epidemiology of physical health issues, and the general recognition of children as a more vulnerable group.

The last emergent theme within the detention period, the negative health impacts of immigration detention, describes the compound downstream effect of detention. Diaz et al. (2023) described the psychological toll as persistent fear and insecurity, a sense of loss and despair, the emergence or worsening of anxiety and depression, and experiences of moral injury. With regard to physical health, the experience of immigration detention was observed to result in the emergence of new physical injuries attributable to violence, the onset of new medical conditions (such as obesity and hyperlipidaemia), as well as difficulties in the management of pre-existing chronic medical conditions (for instance, HIV/AIDS), and

the implications of COVID-19 infection and its sequelae [2]. Notably, all participants had been young and healthy prior to detention, with few pre-migration conditions, yet all experienced deterioration in physical and psychological health [2].

In the UK, most detainees reported high rates of anxiety, distress and depression [67]. Campbell and Steel (2014) found that "mental distress" among asylum seekers encompassed depression, anguish, frustration, stress, hopelessness, and loss of dignity, with all seven participants experiencing these symptoms. Participants reported increased symptoms in detention, linking the environment to mental distress. Professionals noted that restrictive policy conditions in detention contributed to suffering and psychological ill-health for asylum-seekers (ASR) [12]. Mounting hopelessness and demoralisation were recurring in Coffey et al. (2010), with participants struggling to maintain hope and growing increasingly demoralised while detained. For most, this was a gradual deterioration paired with fatigue and loss of vitality, along with reduced ability to act purposefully, seen as a precursor to poor psychological health [3].

Boerma et al. (2022) found that the hopelessness of detention, combined with rumination in the absence of meaningful activity, often re-triggered past traumas, resulting in psychological distress; six participants reported suicidal thoughts, and three had attempted suicide during detention [73]. Isolation in detention is seen as a catalyst, creating a feedback loop of uncertainty, criminalisation, disappointment, and futility in seeking protection. It worsens re-traumatisation and increases anxiety, depression, post-traumatic stress disorder (PTSD) symptoms, and suicidal tendencies [73].

In Canada, Kronick et al. (2015) reported that detained children frequently exhibited oppositional and aggressive behaviours, particularly in those under six years old [69]. Most children exhibited a reduced appetite, with infants requiring more frequent nursing and soothing. Symptoms included separation anxiety, selective mutism, mood disturbance, and PTSD symptoms, indicative of substantial psychological burden related to detention. [69].

Mares & Jureidini (2004) documented very high levels of psychopathology in child and adult asylum seekers, primarily being attributed to traumatic experiences in detention and, for the children, the impact of indefinite detention on their caregivers. All children had at least one parent with a psychiatric illness. Of the 10 children aged 6–17 years, all met criteria for both PTSD and major depressive disorder, with suicidal ideation. Eight (80%), including three pre-adolescents, had made significant self-harm attempts, and seven (70%) showed symptoms of an anxiety disorder. Half reported severe somatic symptoms such as headaches and abdominal pain, and 80% of preschool children exhibited developmental delays or emotional disturbance [74].

Arsenijevic et al. (2018) described compromised psychological well-being and psychosomatic symptoms—headaches, tremors, palpitations—stemming from prolonged stress and cumulative trauma, including detention. Some participants engaged in maladaptive behaviours like substance abuse and self-harm, while others found solace in altruism, religious faith, and family connections [72]. Rothe (2003) similarly observed the effects of cumulative trauma in the interplay among pre-existing trauma, the maritime crossing, and detention experiences, noting extremely high traumatic stress symptom scores in 94% of the boys and 96% of the girls. Behavioural indicators included frequent crying, aggression, nightmares, appetite and sleep disturbances, enuresis, and encopresis, reflecting marked psychological deterioration [71].

### 3.6. Post-detention period

**3.6.1. Theme 4 – Changes in Self-Construct, Relationships and Worldview.** Two subthemes were identified within this theme: fragmentation of identity and changes in self-perception; and erosion of trust and social connectivity.

Witney et al. (2016) measured narrative disruption as a subjective schema ordering the details of an individual's past, present, and future into a meaningful self-construct. The indefinite nature of detention and associated trauma were perceived to fragment this construct. Isolation from the outside world, frequent relocations between facilities, and prolonged solitary confinement were cited as key drivers of deteriorating trust and social connectivity. Solitary confinement also contributed to declining self-confidence and self-regulation, exemplifying reduced agency and adverse changes in self-perception [75].

Passardi et al. (2022) found that some participants experienced shifts in self-construct characterised by guilt and personal culpability, questioning whether their own shortcomings had led to detention [15]. A decline in trust reinforced negative perceptions of self and the world, an inability to experience connection, and difficulty sustaining meaningful friendships. For some, this deterioration was exacerbated by witnessing self-harm and suicide attempts—both during and after detention—often linked to sexual assault experiences. Such events were viewed as "damaging," further entrenching a negative worldview and self-image.

Arsenijevic et al. (2018) reported that detention disempowered ASR by eroding their sense of autonomy and competence, fragmenting their self-perception as capable individuals [72]. Cleveland (2018) described detention as symbolic violence that removes individual agency and choice, including liberty to organise daily life, aspects vital to identity, self-perception, and autonomy. Detention thus disempowers ASR by subjecting them to measures that deprive them of basic rights and freedoms, leading to the fragmentation of their identity and self-concept [68].

Coffey et al. (2010) found that nearly all formerly detained participants experienced significant changes in self-perception, a loss of agency, and reduced sociability. Fractured relationships, social withdrawal, and personality changes were observed as direct consequences of detention – most notably, increased irritability and impatience, distrust of others, and heightened self-doubt, all of which were accompanied by increased rumination [3]. A dampened sense of agency was prominent, as evidenced by impaired capacity for initiative and goal-directed activity, including activities of daily living, language and skill acquisition, and sustaining study or employment, which were all viewed as overwhelming [3].

**3.6.2. Theme 5 – Enduring Impact on Mental and Physical Health.** Two subthemes were identified within this theme: continued psychological sequelae; and physical health deterioration.

Passardi et al. (2022) reported that all but one participant described ongoing psychological sequelae after detention, including depression, emotional numbness, sleep disturbance, obsessive–compulsive behaviours, irritability, and persistent sadness. Physical impacts included new or exacerbated conditions such as skin diseases, renal and bladder calculi, alopecia, ear and eye problems, and Alzheimer's disease. Many attributed these to the combined effects of deprivation, lack of agency, violence, and dehumanisation. Participants described feeling irreparably damaged, hopeless over lost productive years, and aggrieved by a process they perceived as meaningless, with the Australian government seen as culpable [15].

Coffey et al. (2010) likewise found that all participants following their detention reported persistent psychological sequelae due to the ongoing insecurity and prolonged uncertainty regarding their futures and those of their families stemming from the constraints imposed by their visa conditions. These restrictions, coupled with the loss of productive working years while detained, were perceived as acts of injustice affecting all participants post-release. Moreover, participants expressed profound grievance regarding the intrinsic nature of the detention experience, which could not be rationalised and integrated with their suffering, appearing entirely devoid of meaning. As a result, depression, demoralisation, and disturbances in concentration and memory, along with persistent anxiety, were frequently reported. The lasting nature of these detrimental psychological effects was found to persist for several years following their release [3].

For children, immigration detention imposed extreme distress, fear, and functional impairment [69]. Detention was observed to be detrimental to most children after an average duration of 56.4 days. The symptoms incurred did not resolve immediately, leading to sustained adverse psychiatric sequelae well after their release from detention. This phenomenon is partially indicative of cumulative exposure to trauma [70,69].

## 4. Discussion

We report here the key themes emerging from the systematic synthesis of 20 qualitative studies examining the physical and mental health impacts of immigration detention on ASR including children. Themes were grouped into those relating to the detention period and those consequent to detention.

While quantitative systematic reviews have established that immigration detention is associated with significant negative impacts on the physical and mental health of ASR adults and children [27,28,34,77,78], the qualitative evidence presented here expands on this foundation by illuminating the lived experiences and mechanisms underpinning these effects. Whereas quantitative studies quantify the prevalence of disorders and risk factors—often noting elevated rates of PTSD, depression, and anxiety linked to pre-migration trauma and compounded by migration stressors [79,80] —the qualitative data reveal how these stressors are experienced, interpreted, and endured over time. Participants'narratives detail the convergence of pre-existing trauma, the asylum journey, and the conditions of detention—family separation, exposure to violence, discrimination, deprivation of rights, prolonged confinement, isolation, overcrowding, unsanitary environments, and poor-quality food—as catalysts for cumulative psychological and physical harm [2,27,47,48,50,51,56,79,81]. These accounts move beyond symptom prevalence to expose how systemic neglect, erosion of agency, and persistent uncertainty shape mental health trajectories, offering insight into why the harms documented in quantitative research are so pervasive and enduring.

The qualitative data also reveal how the harmful environment is intensified by distress linked to unauthorised arrival status, threats of deportation or repatriation, and fears of renewed persecution [8,82]. Systemic barriers and institutional failings were described as amplifying vulnerability, accelerating mental health decline, and increasing suicidal ideation [2]. In the Australian context, suicide has been reported as the leading cause of premature death in immigration detention [83].

The apparatus and function of immigration detention may be seen as a strategy to deter or remove asylum seekers from the destination country [84]. The literature suggested that a deterrence model predominates, whereby detention centres are used as a mechanism to deter others from seeking protection in industrialised countries, as well as to encourage those who do arrive to return to their country of origin [85]. It is not uncommon for detainees to relinquish their right to appeal their detention as their emotional, financial and health resources become depleted during the prolonged wait time and the associated endemic uncertainty [86]. These factors of protracted uncertainty and reduced self-agency contribute to an environment within detention where human rights violations become possible.

The significant psychological toll of detention is shaped by capriciousness, uncertainty, injustice and moral injury- powerful drivers of traumatic stress [2]. Feelings of powerlessness are strongly correlated with a wide range of mental health symptoms [68]. Indeterminacy in detention length, inactivity, boredom, fear of deportation, family concerns, loneliness, and perceived injustice all contribute to mental distress [68]. ASR who were buoyed by the hope of safety, freedom and a better life and the ability to engage in the realisation of these hopes actively, were buffered against mental distress. Conversely, a sense of powerlessness as a consequence of the immigration detention experience was most strongly correlated with PTSD, depression and anxiety symptoms, with detention sometimes triggering re-traumatisation in trauma survivors [68]. Detention, even for brief periods in relatively adequate conditions, was detrimental to the mental health of ASR [27].

Across studies, protracted and endemic uncertainty was a recurring theme. Waiting is inherent to the asylum-seeking process, which increases uncertainty and has a detrimental effect on well-being [87]. Indefinite delays in refugee status determination foster a sense of powerlessness and reinforce subordination to the arbitrary power of others [68], diminishing social engagement [88]. Persistent low self-esteem and shame inhibit goal-directed activity, reducing productivity and engagement [89]. Chronic uncertainty has been correlated with both psychological and somatoform disorders [87], while prolonged detention and unresolved visa status perpetuate anxiety, depression, hopelessness, and distress [90,91].

### 4.1. Children and families

The UN Convention on the Rights of the Child (UNCRC) stipulates that the detention of children should be a "measure of last resort" (United Nations Office of the High Commissioner for Human Rights [UNHCR], 1989, Article 37. B [92]). Nonetheless, in over 60 countries, children seeking asylum may be subject to immigration detention [93]. Children in detention are particularly vulnerable and are known to experience developmental delays, emotional trauma, and long-term

psychological effects that can impact their development [94]. Detention can be viewed as a form of toxic stress with the potential to interfere with children's capacity to recover from previous trauma and psychopathology [70].

Mares & Jureidini (2004) found that mental health and overall functioning in children deteriorated with prolonged detention [74]. This raises ethical and moral concerns for clinicians bearing witness to a phenomenon that is at odds with international research demonstrating sustained improvement over time among resettled ASR and post-conflict populations [95]. Interactions with mental health clinicians were noted to be humanising and psychologically safe for adolescents, with this hypothesised as serving an environment fostering psychological containment and regulation [71]. Notably, in cases where immigration law took precedence over state health and child protection jurisdictions, clinicians and staff felt morally duty-bound yet immobilised by their inability to effect significant change for children and their parents with severe psychiatric illness and distress [74]. It is important to note that even brief detention appears to be acutely distressing for children, and that constraining and threatening detention environments may serve as an acute stressor, evoking helplessness and fear [69].

### 4.2. Physical health

While somatic illness is acknowledged as salient within populations affected by trauma, only two studies have explored somatic conditions potentially linked to detention in ASR [27,96]. Furthermore, such conditions are frequently conflated with physical health indices, which exhibit considerable overlap and interrelation. Disentangling somatic illness in ASR from physical and mental health indices will be important in understanding the impacts of detention on physical health.

### 4.3. Key Stakeholders- Clinicians and support staff

The staff-detainee relationship was noted to be of paramount importance in de-escalating potential occurrences of self-harm and suicide and addressing mental health concerns. However, detention staff indicated that they could not appropriately recognise and deal with severe mental health distress owing to their lack of training or experience [50,67]. Similarly, volunteer health professionals cited that detention staff were not able to recognise that mental health could deteriorate both in detention as well as in pregnancy and thus, detained pregnant women were not believed when they expressed concerns about their deteriorating mental health. Consequently, appropriate referrals for women to support their mental health were not made [66]. The argument was forwarded that immigration detention facilities are inadequately staffed and resourced, resulting in a systemic failure to implement competently and promptly mandated protocols pertaining to ASR well-being [21].

### 4.4. Strengths and limitations

This qualitative synthesis, to the authors' knowledge, represents the first attempt to synthesise existing qualitative literature on the physical and mental health implications of immigration detention for ASR. A thorough review of qualitative evidence was conducted to incorporate previously overlooked data, thereby offering a more comprehensive understanding of ASR experiences and the health consequences associated with immigration detention. Preliminary thematic frameworks were discussed iteratively within the research team to agree on the emergent themes. A limitation is the complex nature of synthesising studies employing diverse qualitative methodologies; nonetheless, despite this limitation, coherent and integrated findings and themes were successfully extracted, providing unique and significant insights.

The majority of studies concentrate on sample populations situated within the post-detention context rather than on individuals who are detained at the time of recruitment. This observation highlights the possibility of bias in their retrospective accounts of physical and mental health sequelae. This review also underscores the dearth of comprehensive physical health data pertaining to asylum seekers and refugees who have experienced immigration detention. We acknowledge that all authors of this systematic review are healthcare researchers and/or clinicians with a vested interest in the health of ASR. While we do not consider our dedication to advancing healthcare for a marginalised and occasionally politicised

group as a limitation of our work, we do recognise that our professional and personal perspectives inevitably influence our interpretation and critique of the literature.

### 4.5. Clinical implications and future research

There exists a necessity for further research aimed at developing and assessing interventions to assist ASR in managing the significant changes in their lives subsequent to the experience of immigration detention, with a particular emphasis on their physical and mental health. The results of this synthesis may prove beneficial for health professionals, legal practitioners, social workers and policymakers involved in immigration detention, as they can inform the development of future practices and interventions tailored to this population. Additionally, the findings provide insights for future detention policy reviews into the inherent and ameliorable harms and risks encountered by ASRs, both during and following detention.

## 5. Conclusions

This systematic review and thematic synthesis of 20 qualitative studies underscores the significant long-term effects that immigration detention imposes on the psychological well-being of ASR, both during their period of detention and subsequently. While evidence on physical health outcomes is limited, the cumulative consequences of detention highlight the necessity for comprehensive support and healthcare services and associated clinicians and support staff that are tailored to the needs of individuals who have been detained. The findings provide a synthesis of the current literature, illustrating the persistent psychological health implications linked to immigration detention. In contrast, observations regarding physical health have been infrequently documented throughout the studies. This underscores the necessity for host nations to adopt practical and ethical policies aimed at alleviating these adverse effects, even in the context of addressing other policy objectives.

## Supporting information

**S1 Table. Quality Assessment Utilising the Joanna Briggs Institute (JBI) Checklist for Qualitative Research.**
(DOCX)

**S1 Text. Search Strategy.**
(DOCX)

**S1 Data. Characteristics and key findings of the impacts of immigration detention on ASR (n = 20).**
(DOCX)

**S2 Data. Selected Supporting Quotes from Studies.**
(DOCX)

**S1 Checklist. PRISMA Checklist.**
(DOCX)

## Author contributions

**Conceptualization:** Bafreen Sherif.

**Formal analysis:** Bafreen Sherif, Debbie C. Hocking, Susan Rees.

**Methodology:** Bafreen Sherif, Debbie C. Hocking.

**Project administration:** Suresh Sundram.

**Resources:** Suresh Sundram.

**Supervision:** Debbie C. Hocking, Susan Rees, Suresh Sundram.

**Validation:** Debbie C. Hocking, Susan Rees, Letizia M. Affaticati.

**Writing – original draft:** Bafreen Sherif.

**Writing – review & editing:** Debbie C. Hocking, Susan Rees, Letizia M. Affaticati, Suresh Sundram.

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
