## [Decision Letter · Decision Letter 0]

1 Aug 2025

PGPH-D-25-00362

A systematic review of qualitative research on the physical and mental health impacts of immigration detention on asylum seekers and refugees

Dear Dr. Sherif,

Thank you for submitting your manuscript to PLOS Global Public Health. After careful consideration, we feel that it has merit but does not fully meet PLOS Global Public Health’s publication criteria as it currently stands. Therefore, we invite you to submit a revised version of the manuscript that addresses the points raised during the review process. 

We look forward to receiving your revised manuscript.

Kind regards,

Claire J Standley

Academic Editor

Journal Requirements:

Additional Editor Comments (if provided):

In addition to the comments provided through peer review, please consider the additional suggestions and comments provided in the attached file. In particular, consider where text can be streamlined (especially when similar information is already provided in tables/figures), and recommend focusing the Discussion more squarely on where the qualitative data inform and expand on what was already known from quantitative systematic reviews.

Reviewers' comments:

Reviewer's Responses to Questions

**Comments to the Author**

1. Does this manuscript meet PLOS Global Public Health’s publication criteria ? Is the manuscript technically sound, and do the data support the conclusions? The manuscript must describe methodologically and ethically rigorous research with conclusions that are appropriately drawn based on the data presented.

Reviewer #1: Partly

2. Has the statistical analysis been performed appropriately and rigorously?

Reviewer #1: N/A

3. Have the authors made all data underlying the findings in their manuscript fully available (please refer to the Data Availability Statement at the start of the manuscript PDF file)?

Reviewer #1: Yes

4. Is the manuscript presented in an intelligible fashion and written in standard English?

Reviewer #1: Yes

5. Review Comments to the Author

Reviewer #1: PLOS Global Public Health Review: A systematic review of qualitative research on the physical and mental health impacts of immigration detention on asylum seekers and refugees.

I would like to thank the authors for choosing to shed light on the plight of such a population (asylum seekers and immigrants) who are not receiving the attention needed due to the many compounding factors, including the changes in the global context where a population's needs are easily disregarded or deprioritised.

Definitions:

The topic of immigration encompasses health, legal issues, protection, and general social support. This implies that the definitions provided in the articles should conform to standards established by sources that address immigration broadly. Utilising the UNHCR, the International Organisation for Migration, or any relevant sources will enhance the clarity of both the reader's understanding and the article itself, given the multidisciplinary nature of the topic.

The three main terms to clarify in this article are immigrants, asylum seekers, and refugees. It is important for refugees that they cannot move until a green light is granted in some countries; asylum seekers can be at the border or already in the country. Your article focuses on people already in the detention centres, hence the need to clarify this.

Lines 140 and 141 attempt to define asylum seeking; please see the comments above.

Lines 755, 756, and 757 attempt to define a detention centre, which is already useful. My question would be whether this should be on a separate annexe or included in the introduction. It will be important to have these definitions covered in this manuscript.

Length of the process:

The article attempts to clarify how long the asylum-seeking process might take but does not address refugee status. How long does it take to get your results when you are seeking asylum? How long does it take to obtain refugee status?

Clarity:

On lines 98, 99 and 100, the sentence will need to be adjusted with a consideration for another word apart from it (assuming you are referring to the manuscript).

On lines 120, 121 and 122, is it possible to use another word or term that aligns with asylum seeking and being a refugee? The word recruitment sounds like people are happy being in one of the two categories.

Line 62 mentions only two reviewers, under general methods, please include all study authors supporting the literature selection, data extraction and doing the review.

Lines 553 impact on children:

While this is information from the studies, it is important to help the reader understand what you are trying to explain. Why are children not falling under the same category for psychological health and physical health?

Lines 712 – 719 talk about children. I am wondering why you are not putting this information together.

Line 588: Protection is mentioned here, and it is the only time I have not read anything in the paper on protection. Reading your themes, a way to bring back protection, legal, health, and social support would highlight the need for a multidisciplinary approach to this topic.

Flow chart: It should include all the databases searched and the grey literature.

Results: They do not mention anything linked to physical health, as explained under line 165, but the mental health of adults and children. I would have a similar remark for the conclusion, too.

6. PLOS authors have the option to publish the peer review history of their article (what does this mean? ). If published, this will include your full peer review and any attached files.

**Do you want your identity to be public for this peer review?** For information about this choice, including consent withdrawal, please see our Privacy Policy .

Reviewer #1: **Yes: ** Norman Sitali SRN/MIH/MPM

---

## [Decision Letter · Decision Letter 1]

24 Sep 2025

A systematic review of qualitative research on the physical and mental health impacts of immigration detention on asylum seekers and refugees

PGPH-D-25-00362R1

Dear Ms Sherif,

We are pleased to inform you that your manuscript 'A systematic review of qualitative research on the physical and mental health impacts of immigration detention on asylum seekers and refugees' has been provisionally accepted for publication in PLOS Global Public Health.

Best regards,

Claire J Standley

Academic Editor

If further changes are required by the editorial office, please consider reviewing the text carefully for mentions of "Asylum Seeker and Refugees" after it is first introduced with the acronym ASR, and replace with ASR the rest of the time (i.e. line 187, 262, 927 etc.).

Reviewer Comments (if any, and for reference):

Reviewer's Responses to Questions

**Comments to the Author**

1. If the authors have adequately addressed your comments raised in a previous round of review and you feel that this manuscript is now acceptable for publication, you may indicate that here to bypass the “Comments to the Author” section, enter your conflict of interest statement in the “Confidential to Editor” section, and submit your "Accept" recommendation.

Reviewer #1: All comments have been addressed

2. Does this manuscript meet PLOS Global Public Health’s publication criteria ? Is the manuscript technically sound, and do the data support the conclusions? The manuscript must describe methodologically and ethically rigorous research with conclusions that are appropriately drawn based on the data presented.

Reviewer #1: Yes

3. Has the statistical analysis been performed appropriately and rigorously?

Reviewer #1: N/A

4. Have the authors made all data underlying the findings in their manuscript fully available (please refer to the Data Availability Statement at the start of the manuscript PDF file)?

Reviewer #1: Yes

5. Is the manuscript presented in an intelligible fashion and written in standard English?

Reviewer #1: Yes

6. Review Comments to the Author

Reviewer #1: Dear Authors, thank you for the adjustment to the manuscript based on the comments provided in the last submitted version. I only have two points, one is minor and one is an important one:

1. ASR has been explained as an abbreviation in full, which is good for the readers, this means the rest of the document should use ARS not Asylum Seeker and Refugees (ASR) all the time. Check line 187, 262, 927 etc.

2. Line 335, here you mention that the methodology quality was not considered for the article included in the review? The overall strength and trustworthiness of a systematic review depend heavily on the quality of the studies included. A strong methodology ensures that the findings are valid and reflect the true partners or themes in the data.

7. PLOS authors have the option to publish the peer review history of their article (what does this mean? ). If published, this will include your full peer review and any attached files.

**Do you want your identity to be public for this peer review?** For information about this choice, including consent withdrawal, please see our Privacy Policy .

Reviewer #1: **Yes: ** Norman Sitali RN/MPH/MIH
